# Hippocampal pyramidal cells of the CA1 region are not a major target of the thalamic nucleus reuniens

Lilya Andrianova[1,2,3], Paul J. Banks[4], Clair A. Booth[4], Erica S. Brady[3], Gabriella Margetts-Smith[3], Shivali Kohli[3], Jonathan Cavanagh[2], Zafar I. Bashir[4], Chris J. McBain[5], Michael T. Craig[1,3,5]*

1 School of Psychology & Neuroscience, College of Medical, Veterinary and Life Sciences, University of Glasgow, Glasgow, United Kingdom, 2 School of Infection & Immunity, College of Medical, Veterinary and Life Sciences, University of Glasgow, Glasgow, United Kingdom, 3 Institute of Biomedical and Clinical Science, University of Exeter Medical School, Exeter, United Kingdom, 4 School of Physiology, Pharmacology and Neuroscience, University of Bristol, Bristol, United Kingdom, 5 Program in Developmental Neurobiology, Eunice Kennedy Shriver National Institute of Child Health and Human Development, National Institutes of Health, Bethesda, Maryland, United States of America

* mick.craig@glasgow.ac.uk

## Abstract

The prefrontal—hippocampal—entorhinal system is perhaps the most widely-studied circuit in cognitive and systems neuroscience, due to its role in supporting cognitive functions such as working memory and decision-making. Disrupted communication within this circuit is a key feature of disorders such as schizophrenia and dementia. Nucleus reuniens (NRe) is a midline thalamic nucleus that sits at the nexus of this circuit, linking these regions together. As there are no direct projections from prefrontal cortex to hippocampus (HPC), the accepted model is that the NRe mediates prefrontal drive of hippocampal activity, although these connections are poorly defined at the cellular and synaptic level. Using *ex vivo* optogenetics and electrophysiology in both mice and rats, alongside monosynaptic circuit-tracing, we sought to test the mechanisms through which NRe could drive hippocampal activity. Unexpectedly, we found no evidence that pyramidal cells in CA1 receive input from NRe, with midline thalamic input to HPC proper appearing selective for GABAergic interneurons. In other regions targeted by NRe, we found that pyramidal cells in prosubiculum and subiculum received synaptic inputs from NRe that were at least an order of magnitude weaker than those in prefrontal or entorhinal cortices. We conclude that, contrary to widely-held assumptions in the field, hippocampal pyramidal cells are not a major target of NRe.

## Introduction

Numerous aspects of cognition, such as memory [1] and decision-making [2], require the communication of information between prefrontal cortex (PFC), hippocampus (HPC), and entorhinal cortex (EC). The thalamic nucleus reuniens (NRe) sits at the

**Data availability statement:** All data are contained in the paper and Supporting information files, and the R code required for analysis (S6 Fig) has been included as a supplementary file.

**Funding:** This work was supported by Biotechnology and Biological Sciences Research Council (www.ukri.org/councils/bbsrc/) grant BB/P001475/1 (awarded to M.T.C.) and an NIH intramural award (awarded to C.J.M.). This work was also supported by a Wellcome Trust (https://wellcome.org/) Joint Investigator Award 206401/Z/17/Z (awarded to Z.I.B.), and Biotechnology and Biological Sciences Research Council (www.ukri.org/councils/bbsrc/) grants BB/X000915/1 and BB/L001896/1 (awarded to Z.I.B. and P.J.B.). Salary support from S.K. was provided by Alzheimer's Research UK (www.alzheimersresearchuk.org/) Interdisciplinary research grant ARUK-IRG2017B-4 (M.T.C.). G.M.S. and E.S.B. were both GW4 BioMed doctoral training program students funded by the Medical Research Council (www.ukri.org/councils/mrc/) grant MR/N0137941/1. This work was also supported by the Inger and George Simpson Biological Psychiatry Scholarships (Endowment to University of Glasgow; awarded to J.C.). The funders had no role in study design, data collection and analysis, decision to publish, or preparation of the manuscript.

**Competing interests:** The authors have declared that no competing interests exist.

**Abbreviations:** ABC, avidin–biotinylated HRP complex; ACC, anterior cingulate cortex; ACUC, animal care and use committee; AP, anterior-posterior; dCA1, dorsal CA1; EC, entorhinal cortex; EPSC, excitatory post-synaptic current; HPC, hippocampus; IL, infralimbic cortex; MO, medial orbital cotex; NGF, neurogliaform cell; NMDA-R, N-methyl-D-aspartate receptor; NRe, nucleus reuniens; PB, phosphate buffer; PFC, prefrontal cortex; PL, prelimbic cortex; RMP, resting membrane potential; SL-M, stratum lacunosum-moleculare; vCA1, ventral CA1.

nexus of this circuit and is implicated in conditions such a schizophrenia and epilepsy [3]. This circuit includes both direct connections between the cortical and hippocampal regions, as well as indirect routes via the thalamus, making interpretation of the function of any individual component particularly challenging. Synchronous oscillations occur between the PFC and HPC at theta frequency during decision-making [4,5], between the EC and PFC at theta frequency during associative learning [6], and between the EC and HPC at theta and gamma frequencies during spatial learning [7]. There is, however, an *anatomical anomaly* within this circuitry: despite the functional importance of harmonized activity, the PFC does not project directly to HPC [see 8 for our recent confirmation]. The accepted view of this circuit is that the NRe directly mediates communication between PFC and HPC [e.g., 9, 10], which implicates the NRe in cognitive functions such as working memory [11,12], and leading many to believe that the main function of the NRe is to mediate prefrontal control over the HPC. However, NRe inactivation or lesion studies often fail to provide strong evidence for a role in spatial memory acquisition [13].

Anatomically, the NRe forms reciprocal connections with the PFC and subiculum, while uniquely for the thalamus, it also sends direct efferents to CA1; these efferents terminate alongside EC axons in stratum lacunosum-moleculare (SL-M) [14,15]. Hippocampal return projections to the NRe arise entirely from the subiculum, with neither dorsal nor ventral CA1 (vCA1) forming monosynaptic connections with neurons in NRe [16]. The NRe sends projections to the EC [17], but these are effectively not reciprocated [18] with perhaps only 2% of superficial the EC neurons projecting to NRe [16]. Functionally, although electrical stimulation in the NRe fails to elicit spiking in the CA1 pyramidal layer *in vivo* [19], it has been widely assumed that the NRe targets CA1 pyramidal cells [11]. This assumption is reasonable as we are unaware of any cortical region in which thalamic inputs do *not* target pyramidal cells. Surprisingly, despite the NRe's important role in goal-directed spatial navigation [20] and working memory [21], the NRe projections to CA1 have remained poorly defined. We previously reported that both NRe and entorhinal fibers terminate in SL-M of CA1 where they target neurogliaform cells [22]. Optogenetic stimulation of the NRe inputs to these neurogliaform cells elicits monosynaptic EPSCs that are defined by large NMDA receptor-mediated components.

Here, we sought to determine the nature of NRe inputs onto CA1 pyramidal cells. We hypothesized that a similarly large NMDA-R mediated component in pyramidal cells to what we observed in neurogliaform cells could underlie observation that NRe-fEPSPs are greatly increased in CA1 during EC input coactivation [23]. However, we found no evidence of direct monosynaptic projections from the NRe to CA1 pyramidal cells, and only evidence of weak inputs to pyramidal cells in other areas of the hippocampal formation.

## Results

### NRe projections to hippocampal formation

We carried out stereotaxic injection of AAVs to express the light-sensitive channelrhodopsin2 variants Chronos or Chrimson [24] into the NRe and prepared *ex vivo* brain

slices in either coronal or horizontal planes to carry out patch-clamp recordings targeting primarily PFC and dorsal HPC, or EC and ventral HPC, respectively (example distribution of NRe axons in HPC shown in S1 Fig). We subdivided the HPC into CA1, prosubiculum, and subiculum, using the Lorente de Nó boundaries [25]. We denoted the boundary of CA1 to be where the dense, ordered nature of stratum pyramidale ended, with prosubiculum continuing in an area with a slight laminar-like organization in the superficial region (see Fig 2 and particularly figure 11 of the Lorente De Nó paper), with subiculum beginning when no obvious laminar structure was apparent through DIC optics. The boundaries that we used correspond well with those reported more recently [26], and are shown in Figs 1A (ventral HPC) and 2A (dorsal HPC).

As the NRe sends a substantially denser projection to ventral over dorsal HPC [14], we first carried out patch-clamp recordings in vCA1. We found that optogenetic activation of NRe axons in CA1 failed to elicit post-synaptic EPSCs in vCA1 pyramidal cells (Figs 1B and 3A). We observed the same result in dorsal CA1 (dCA1) (Figs 2B and 3A). In each slice tested, we only recorded a pyramidal cell as nonresponsive when we could evoke a post-synaptic response in neurogliaform cells (NGFs) that, with their high NRe input probability, acted as positive controls (Figs 1D–1F for vCA1 and 2C–2E for dCA1; summary data Fig 3A). We found that there was no difference in the magnitude of NRe-EPSC on neurogliaform cells when evoked using either Chronos or Chrimson (S2 Fig). Additionally, we found no differences in NRe input probability or synaptic strength between dorsal and ventral HPC, nor did we observe differences in NRe-EPSC magnitude between MGE- and CGE-derived neurogliaform cells (S3 Fig).

Unlike CA1, we found that NRe projections to other parts of the HPC (prosubiculum and subiculum; Figs 1G–1K for vCA1 and 2F–2H for dCA1) did indeed target principal cells although the synaptic strength of these inputs was low (−13.32 ± 2.7 pA for subiculum; summary data in Fig 3), suggesting that NRe projections to hippocampal principal cells are either absent (CA1) or subthreshold (prosubiculum and subiculum Fig 3), and unlikely to elicit firing. NRe projections to CA1 neurogliaform cells had a comparable magnitude of the AMPA-receptor-mediated component of the EPSC (Fig 3C), but the higher input resistance of inhibitory interneurons combined with the larger NMDA–AMPA ratio (Fig 3E) would suggest that NRe could drive neurogliaform cells to spike, perhaps during periods of ongoing network activity.

## NRe projections to nonhippocampal areas

We also investigated the connectivity of the NRe to the cortical target regions, prefrontal (Fig 4) and entorhinal cortices (Fig 5). Similarly to the hippocampal region, the nonresponding cells were only taken into account if the same slice contained a positive control. Directly comparing NRe-evoked EPSC responses in cortical and hippocampal neurons, we found that principal cells in the MEC and PFC pyramidal cells displayed significantly larger AMPA receptor-mediated EPSCs than subiculum pyramidal cells or CA1 NGFs (Fig 3A, 3B, and 3D). These data would suggest that, from a functional perspective, the EC and PFC, and not the HPC, may be the principal targets of thalamic NRe.

When looking at the PFC overall, the majority of the pyramidal cells were responsive to optogenetic stimulation of NRe axons (Figs 3A and 4A). We saw a similar result in the medial EC (Figs 3A and 5A). We saw no difference in NRe-EPSC amplitude in the PFC when comparing NRe input between medial orbital, infralimbic, prelimbic, and anterior cingulate areas (Fig 4B–4D). While there are at least two main principal cell subtypes in the EC [27–29], we did not parse EC neurons further into stellate or pyramidal cells, so we cannot comment on whether the large variability in NR-EPSC magnitude observed (Fig 5D) corresponded to different subtypes of neuron. Nevertheless, these data confirm that NRe forms strong synaptic inputs onto principal neurons in both prefrontal and entorhinal areas.

To exclude the possibility that a lack of electrophysiological response from NRe axons onto CA1 pyramidal cells was unique to mouse, we undertook an independent replication of this finding in a different laboratory using rats. We found that 0/9 dCA1 pyramidal cells and 0/17 vCA1 pyramidal cells in rat HPC displayed post-synaptic responses in response to optogenetic stimulation of NRe axons, but that more than half of all interneurons in SL-M did indeed receive a response (S4 Fig). Importantly, PFC pyramidal neurons from the same preparations responded robustly to NRe input with similar currents to those seen in mouse [30].

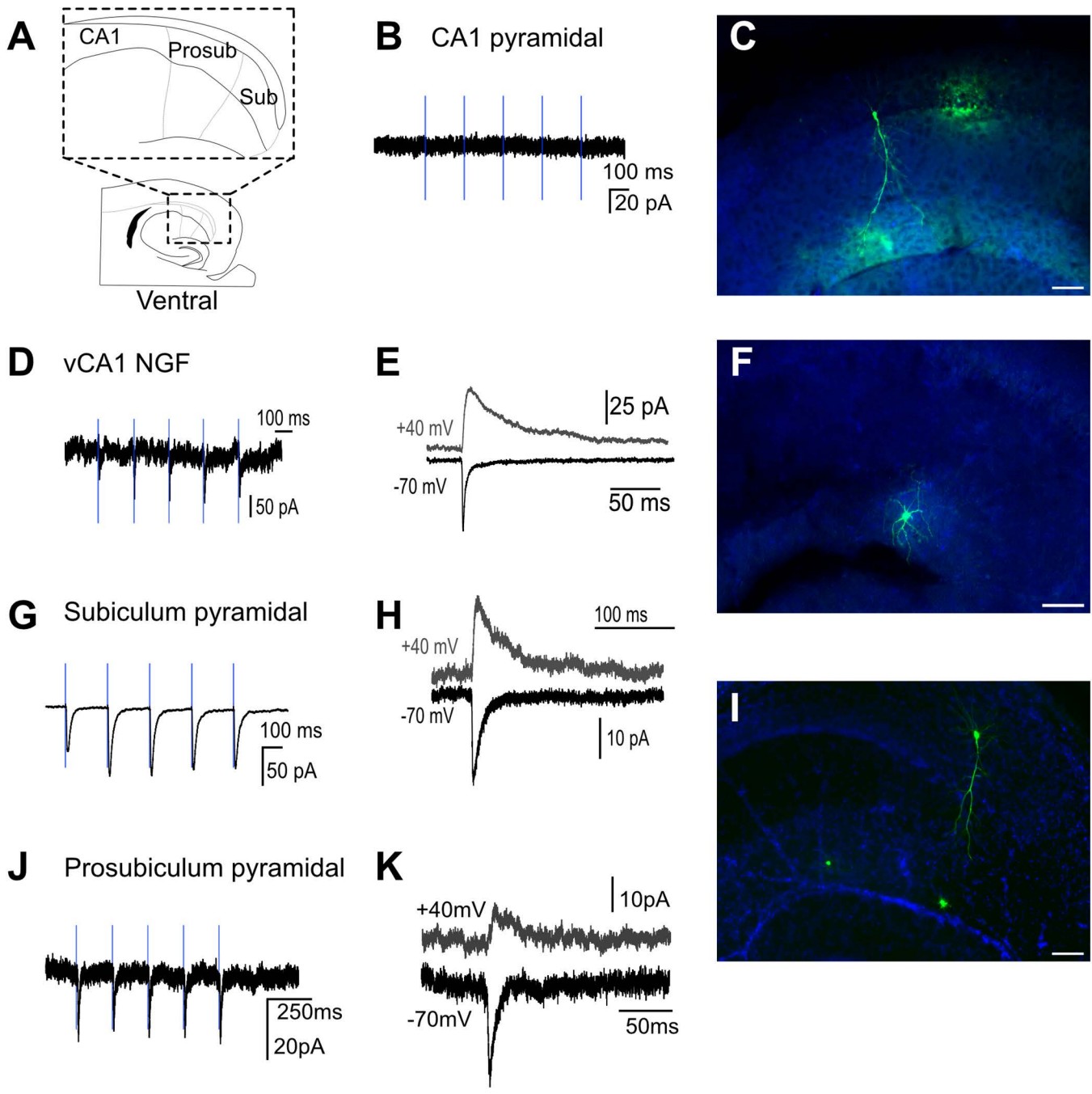

**Fig 1. Prosubiculum pyramidal cells, unlike those in ventral CA1, do receive an input from nucleus reuniens.** Schematic representation of boundaries between dorsal CA1, prosubiculum, and subiculum in horizontal orientation. **A**, CA1/prosubiculum boundary denoted as region where additional pyramidal cell layers arise 'below' the tightly-packed layer that is a continuation of CA1 stratum pyramidale, after Lorento de No (1934). **B**, No response recorded from dCA1 pyramidal cell. **C**, *Post hoc* recovery of a nonresponsive pyramidal cell in dCA1. **D**, Representative response trace from a ventral NGF. **E**, AMPA and NMDA receptor-mediated components of a responsive ventral NGF cell. F, A *post hoc* recovery from a ventral NGF cell. **G**, Example trace from responsive pyramidal cell in dorsal subiculum. **H**, AMPA and NMDA receptor-mediated components of a responsive pyramidal cell from subiculum. **I**, *Post hoc* recovery of a responsive pyramidal neuron in prosubiculum. **J**, Example trace from a responsive pyramidal cell in dorsal prosubiculum. **K**, AMPA and NMDA receptor-mediated components of a responsive pyramidal cell from prosubiculum. Scale bar 100 μm.

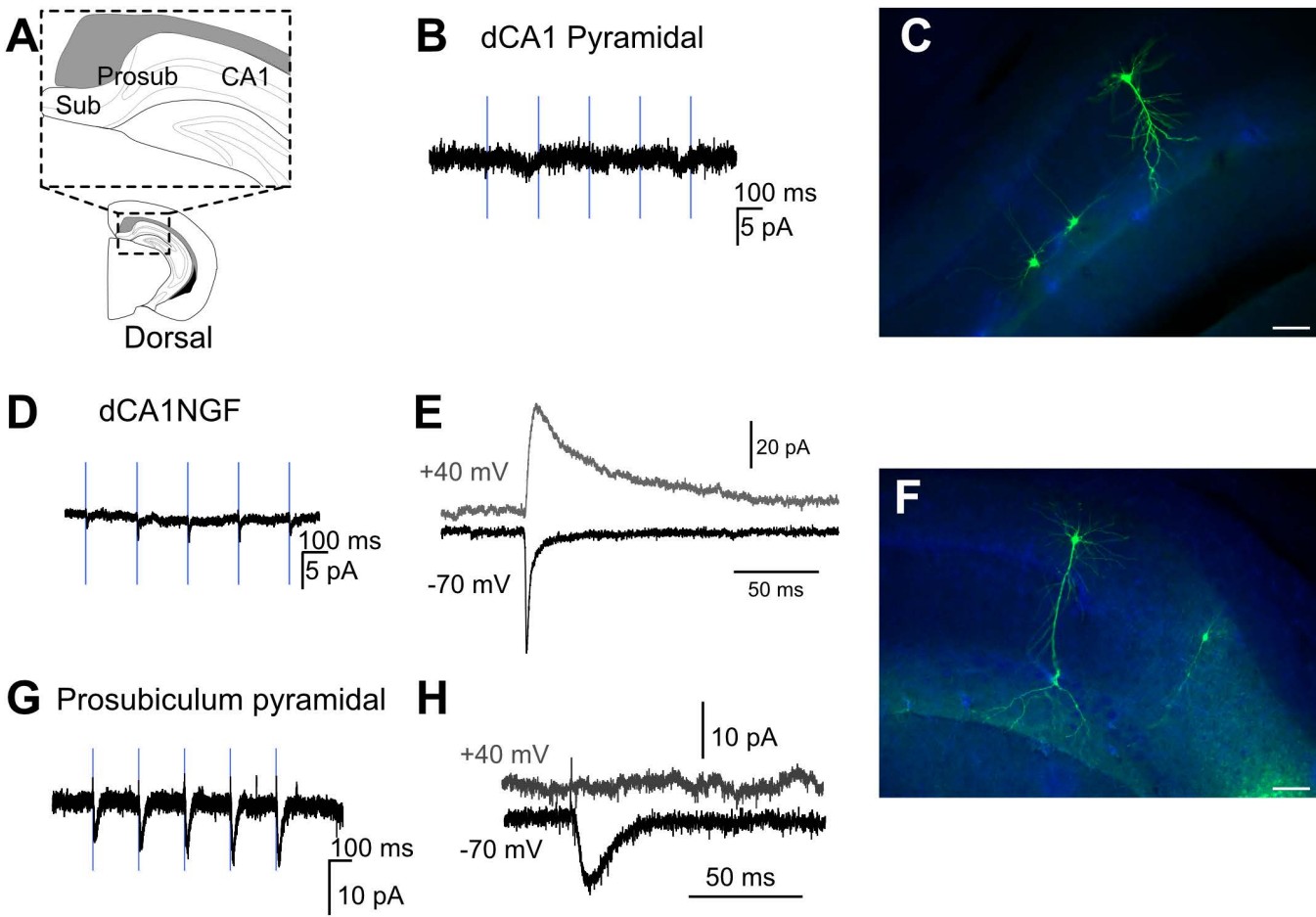

**Fig 2. Prosubiculum pyramidal cells, unlike those in dorsal CA1 (dCA1), do receive an input from nucleus reuniens. A,** Schematic representation of boundaries between dCA1, prosubiculum, and subiculum in coronal orientation. **B,** Example trace from a nonresponsive pyramidal neuron from dCA1 **C,** *Post hoc* recoveries of two neurogliaform cells and a nonresponding CA1 pyramidal cell. **D,** A representative trace from a responsive dorsal NGF. **E,** AMPA and NMDA receptor-mediated components of a responsive NGF from dCA1. **F,** *Post* hoc recoveries of two pyramidal cells, the nonresponding CA1 pyramidal cell (labeled by arrowhead) and responding pyramidal cell in the prosubiculum (labeled with rounded arrowhead) as well as a dorsal NGF. **G,** Example trace from a responsive pyramidal neuron from dorsal prosubiculum. **H,** AMPA and NMDA components of a responsive pyramidal neuron from dorsal prosubiculum. Scale bar 100 μm.

## Monosynaptic retrograde tracing from CA1 pyramidal cells

Although our patch-clamp electrophysiological recordings failed to find evidence of somatic EPSCs mediated by NRe inputs in CA1 pyramidal cells, given that NRe ESPCs in neurogliaform cells and subiculum pyramidal cells have a small magnitude (Fig 3), we could not exclude the possibility that inputs were present but undetectable due to dendritic filtering of the NRe inputs arriving in the distal region of apical dendrites, despite having found no evidence of silent NMDA-R only synapses. Consequently, we carried out rabies-assisted monosynaptic circuit-tracing in *Emx1-cre* mice that conditionally expressed the avian TVA receptor only in pyramidal cells (see Materials and methods and Fig 6). In our experimental approach, as TVA was expressed genetically in pyramidal cells, it was possible for rabies viruses to locally transduce pyramidal cells in CA1. However, we used a pseudotyped glycoprotein-deficit SAD B19 version of the rabies virus that is unable to replicate and spread from transduced neurons to express mCherry. Prior to rabies injection, we used an AAV helper virus to express GFP and the rabies glycoprotein (see Materials and methods and S5 Fig). Thus, in our experiment,

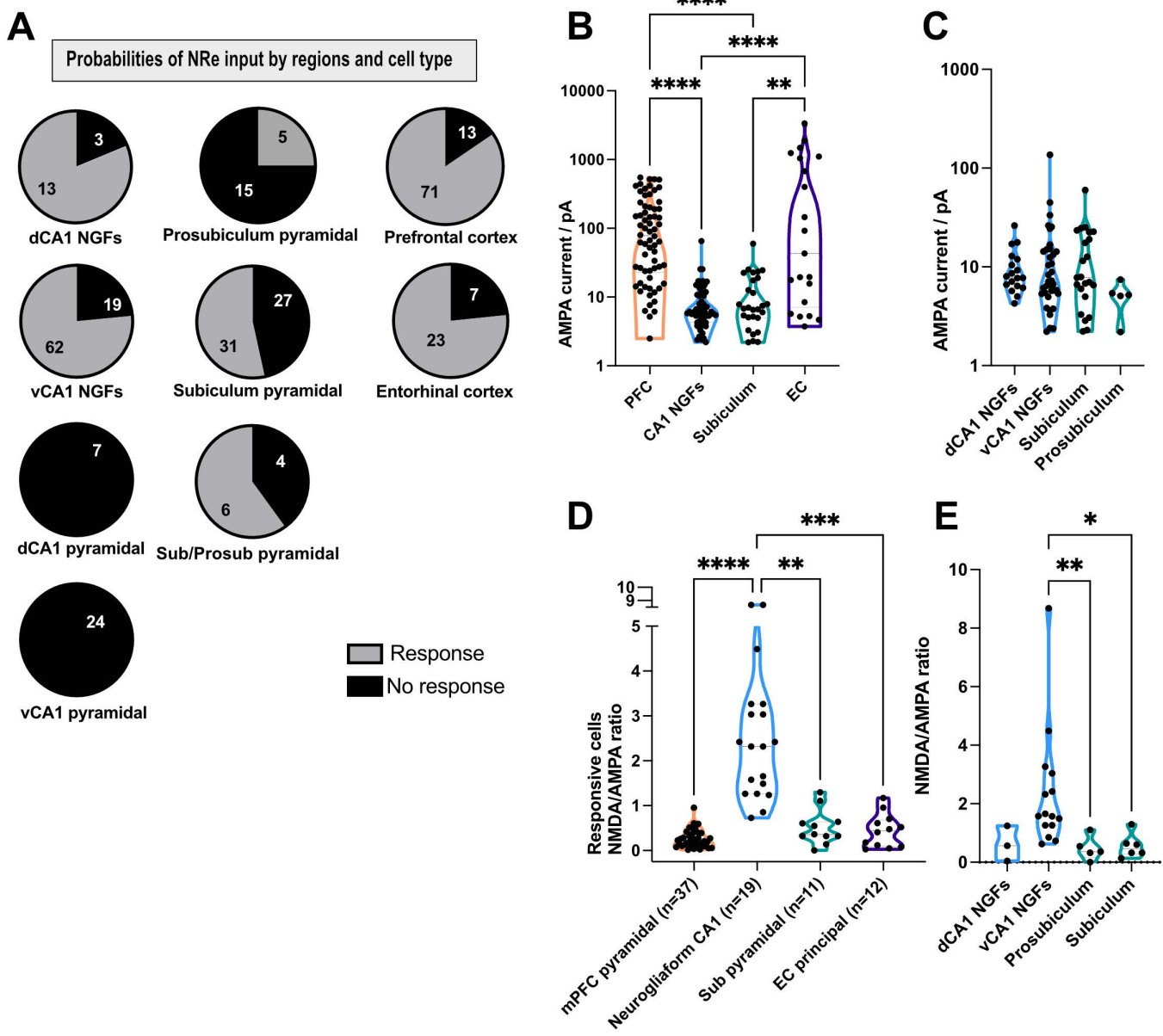

**Fig 3. Summary data for all different cell types in the hippocampal formation and cortical areas. A,** Nucleus reuniens (NRe) input probability for cells patched in hippocampus: dorsal NGFs (81.3%) and ventral NGFs (76.5%), dorsal (0%) and ventral (0%) CA1 pyramidal cells, prosubiculum (66.7%), subiculum (46.6%), and undetermined prosubiculum/subiculum (60%) pyramidal neurons, prefrontal cortex (84.5%), and entorhinal cortex (EC) (76.7%). **B,** AMPA currents vary across different NRe target regions, with average AMPA currents being −142±19 pA for PFC pyramidal cells, −9.1±1.2 pA for NGFs, −12±2.3 pA for pyramidal cells in subiculum, and −551±190 pA for principal cells in the EC. Note logarithmic scale. Kruskal–Wallis test (with Dunn's multiple comparison test) showed significant differences between CA1 NGFs vs. EC ($p<0.0001$), CA1 NGFs vs. PFC ($p<0.0001$), EC vs. Subiculum ($p=0.0014$), and Subiculum vs. PFC ($p<0.0001$). **C,** AMPA currents across different regions with the average being −9.88±1.3 pA (dCA1 NGFs), −14.33±3.7pA (vCA1 NGFs), −13.32±2.7 pA (Subiculum pyramidal cells), −5.06±0.8 pA (Prosubiculum pyramidal cells), no significant difference found using a one-way ANOVA with Tukey's multiple comparisons test. **D,** NMDA/AMPA ratios of responding cells in different regions, average ratios were 1.1±0.26 (dCA1 NGFs), 3.8±1.7(vCA1 NGFs), 0.47±0.18 (Subiculum pyramidal cells), 0.55±0.17 (Prosubiculum pyramidal cells). Using a Kruskal–Wallis test with Dunn's multiple comparisons test significant differences were found for NGFs vs. Subiculum pyramidal cells ($p=0.0047$), EC principal vs. NGFs ($p=0.0003$), and PFC pyramidal cells vs. NGFs ($p<0.0001$). **E,** Kruskal–Wallis test with Dunn's test for multiple comparisons, vNGFs cells have a significantly higher NMDA/AMPA ration when compared to subiculum and prosubiculum pyramidal neurons ($p=0.0206$ and $p=0.0074$, respectively). $N$=cell. These data can be found in the S1 Data.

PLOS Biology

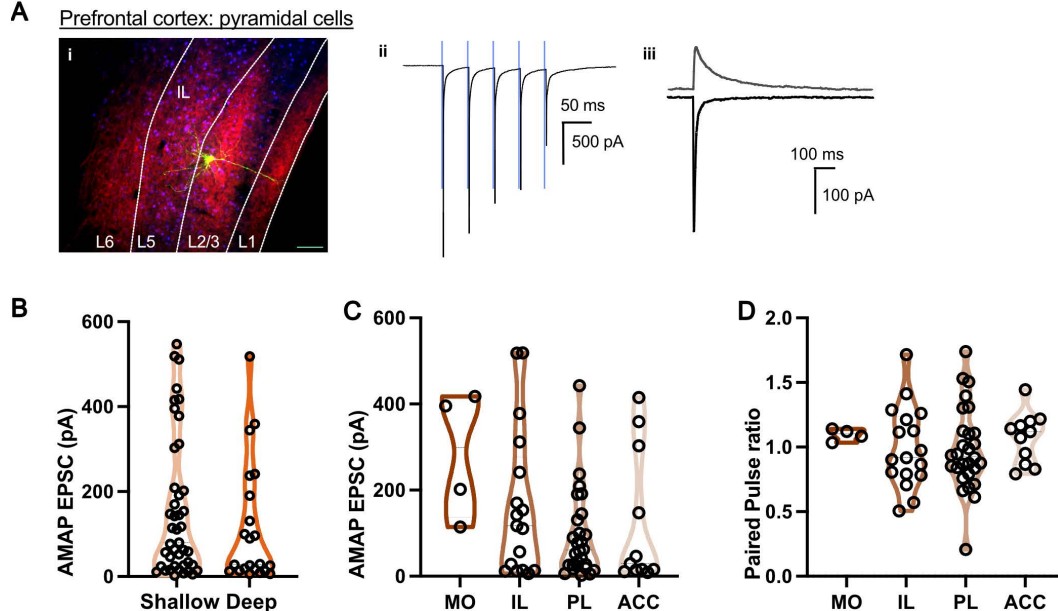

**Fig 4. Nucleus reuniens (NRe)-EPSCs are consistent across all subdivisions of the prefrontal cortex (PFC). A,** A representative *post hoc* recovery of a pyramidal neuron in shallow layers of PFC (i, scale bar 100-μm), optogenetic response trace (ii), AMPA and NMDA currents (iii). **B,** AMPA-R mediated NRe-EPSC did not vary significant between deep (layer 2 to 3) and shallow (layer 5 to 6) pyramidal cells in PFC (shallow *vs.* deep: 155±25 pA (*n*=43) vs. 114±30 pA (*n*=22); *p*=0.232, Mann–Whitney test). **C,** AMPA-R mediated NRe-EPSC did not vary between different subdivisions of PFC (MO vs. IL vs. PL vs. ACC: 282±74 pA (*n*=4) vs. 165±42 pA (*n*=17) vs. 98±20 pA (*n*=28) vs. 147±56 pA (*n*=9 cells); *p*=0.46, one-way ANOVA). **D,** Paired pulse ratio of AMPA-R mediated NRe-EPSC did not vary between different subdivisions of PFC (MO vs. IL vs. PL vs. ACC: 1.1±0.02 vs. 1.0±0.08 vs. 0.98±0.06 vs. 1.1±0.06; *p*=0.15, one-way ANOVA). *N*=cells. These data can be found in the S1 Data.

double-labeled GFP and mCherry cells in CA1 or the subiculum are starter cells from which the rabies virus could retrogradely spread a single synapse, while mCherry-only cells in CA1 could be either presynaptic neurons or transduced with the pseudotyped vector. mCherry-only cells in other brain regions would be those presynaptic to CA1 pyramidal cells. Some data from this tracing experiment have previously been published elsewhere [8].

We failed to find evidence of monosynaptic NRe inputs to pyramidal cells in either dorsal (Fig 6A and 6E; *n*=6 mice) or ventral (Fig 6F and 6I, *n*=4) CA1 but, as expected, we saw consistent retrograde labeling in entorhinal cortices, medial and lateral septa and subiculum (we previously published some of this dataset in [8], confirming prior findings from Sun and colleagues [31]). We only observed retrograde labeling of NRe neurons when injections into ventral hippocampal formation included starter cells in both vCA1 and ventral prosubiculum/ subiculum (Fig 6J–6L; *n*=3; all mice had retrogradely-labeled cells in NRe). The boundaries between the CA1, prosubiculum, and subiculum regions in ventral HPC in horizontal plane are shown in Fig 6M. Quantification of double-labeled starter cells, estimated overall number of starter cells as well as viral spread data is shown in Fig 6N–6P, and reproduced from our previous work [8]. To exclude the possibility that *Emx1* is not ubiquitously expressed throughout the HPC, we carried out a secondary analysis of RNAseq datasets publicly available from [32] and found that *Emx1* is indeed expressed throughout CA1 (S6 Fig).

## Discussion

Anatomical studies of NRe connectivity have led many to believe that the main function of the NRe is to mediate prefrontal control over the HPC [e.g., 11]. However, our findings challenge this core assumption. We found that the magnitude of synaptic currents driven by NRe axons is at least an order of magnitude greater in the EC and PFC than in either CA1

PLOS Biology

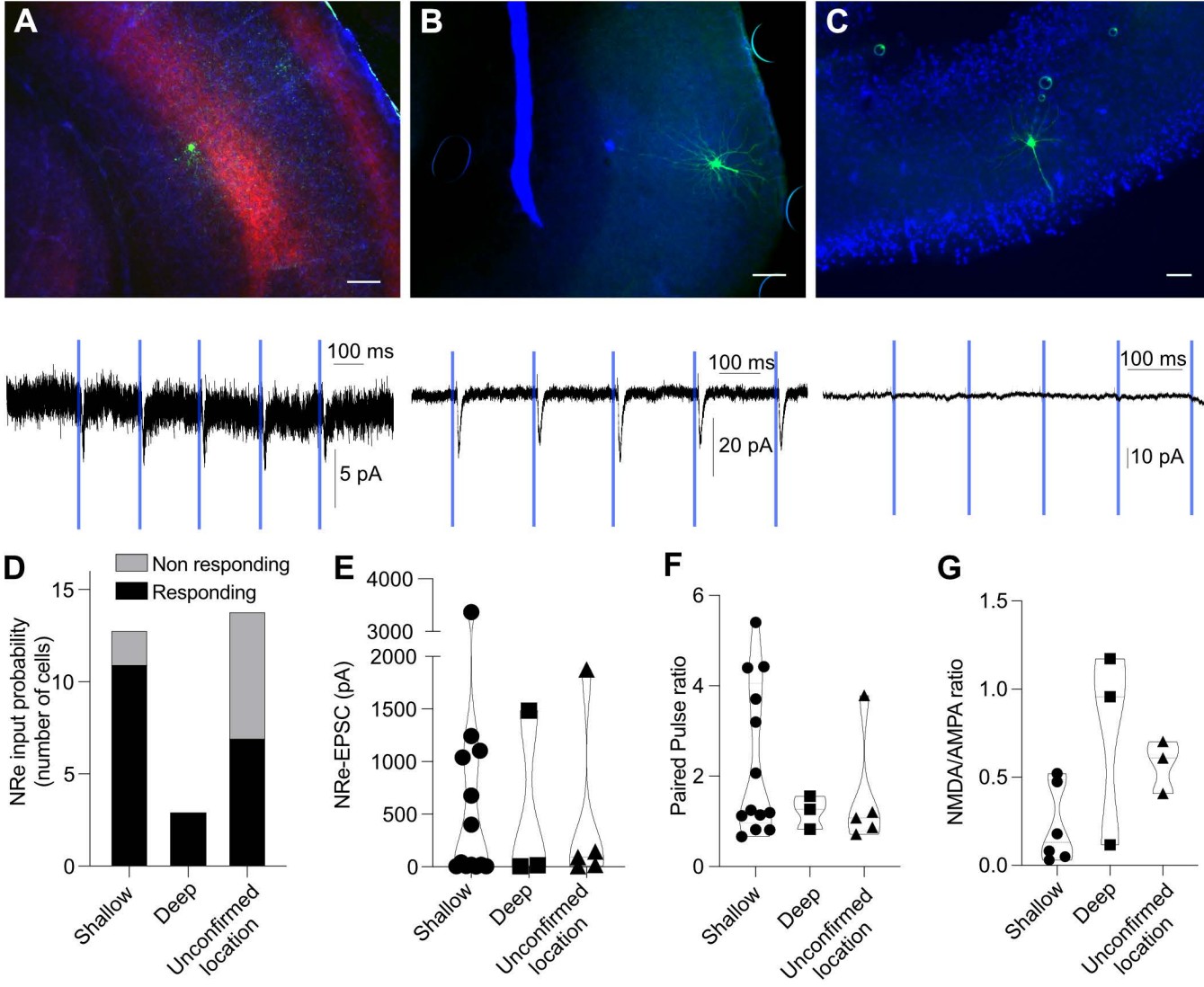

**Fig 5. Nucleus reuniens (NRe)-EPSCs are consistent across shallow and deep layers of entorhinal cortex. A,** *Post hoc* recovery of layer 5 mEC pyramidal neuron (i) and optogenetic response trace (ii). **B,** *Post hoc* recovery of layer 2 mEC stellate cell (i) and optogenetic response trace (ii). **C,** *Post hoc* recovery of layer 2 mEC pyramidal neuron (i) and lack of response to optogenetic stimulus (ii). **D,** Input probability of mEC neurons. **E,** AMPA-R mediated NRe-EPSCs are not significantly different in shallow vs. deep layers of mEC. **F,** NMDA/AMPA ratios in responsive neurons in shallow and deep layers of mEC are not significantly different. **G,** Paired pulse ratio in responsive neurons in shallow and deep layers of mEC is not significantly different. These data can be found in the S1 Data.

or subiculum. Furthermore, we have found that pyramidal cells in CA1 appear *not* to receive monosynaptic inputs from the NRe (determined using both anterograde optogenetic circuit-mapping and monosynaptic retrograde rabies tracing). It appears that the NRe preferentially targets inhibitory interneurons in CA1 and pyramidal cells in prosubiculum.

## Do CA1 pyramidal cells receive input from NRe?

Our data cannot entirely exclude the possibility of a monosynaptic projection from NRe to CA1 pyramidal cells, but if the connection does exist then it must be very sparse. Unlike the HPC proper, we found that pyramidal cells in both

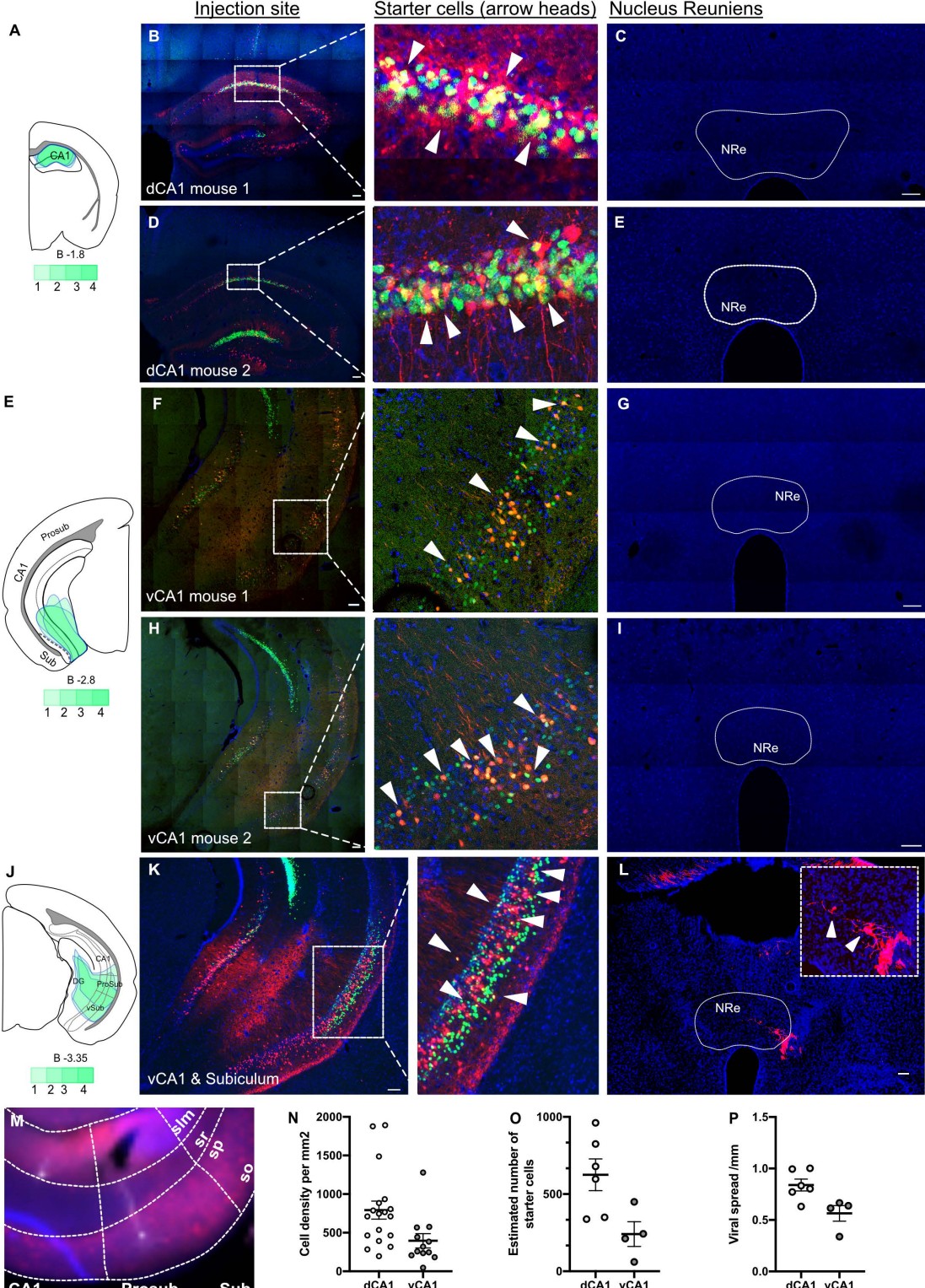

**Fig 6. Monosynaptic retrograde tracing from hippocampus proper shows no cells originating from nucleus reuniens (NRe), unless the starter cells are present in subiculum region. A,** Injection site for dorsal CA1 and viral spreads for the helper virus (green cells). **B** and **D,** Injection site of a dCA1 injection: green cells contain helper virus and red cells are pseudotyped rabies virus positive and starter cells (yellow) shown closer on the inset,

shown by arrowheads. **C** and **E,** No cells found in any thalamic section containing NRe. **E,** Injection site for ventral CA1 (vCA1) and viral spreads for the helper virus (green cells). **F** and **H,** Injection site of a vCA1 injection: green cells contain helper virus and red cells are pseudotyped rabies virus positive and starter cells (yellow) shown closer on the inset, shown by arrowheads. **G** and **I,** No cells found in any thalamic section containing NRe. **J,** Injection site for vCA1 & subiculum and viral spreads for the helper virus (green cells). **K,** Injection site of a vCA1 injection with virus spreading to subiculum: green cells contain helper virus and red cells are pseudotyped rabies virus positive and starter cells (yellow) shown closer on the inset, shown by arrowheads. **L,** Cells and fibers seen in NRe as well as the fornix. **M,** A post hoc recovery of pyramidal cell in prosubiculum and neurogliaform cell in SL-M layer of CA1 in horizontal slice orientation; boundary between CA1 and prosubiculum can be determined by reduction in SR layer thickness and reduction in canonical cell density in SP layer and the boundary between prosubiculum and subiculum is where SR layer almost disappears. **N,** Cell density showing number of cells containing the pseudo-typed rabies virus in dorsal and vCA1 regions. **O,** Estimated number of starter cells, calculated from cell density and viral spread. **P,** Viral spread distance in dorsal and vCA1 regions. For more details on retrograde monosynaptic tracing experiment data see [8]. These data can be found in the S1 Data.

prosubiculum and subiculum do indeed receive an input from NRe. The lack of input to pyramidal cells in our study was very specific to CA1; pyramidal cells in prosubiculum, identified using Lorente de Nó anatomical definitions [25], did receive direct input from NRe. While some regard the prosubiculum as the distal region of CA1, recent transcriptomic studies confirm that it is genetically distinct from both CA1 and subiculum [33], and our findings support this. The regions that we recorded from in the prosubiculum were clearly outwith the boundaries of CA1, where the dense organization of neurons in stratum pyramidale ended. No pyramidal neuron recorded within the more highly-ordered CA1 responded to optogenetic stimulation of thalamic axons, while we always used neurogliaform cells as positive controls. It has been shown that the NRe selectively innervates the pyramidal cells in the ventral subiculum that target the nucleus accumbens and lateral habenula, whilst avoiding those that target the PFC [34].

When we first reported these findings as a preprint in late 2021 [35], the results appeared controversial. Indeed, while a recent publication reported a direct monosynaptic projection from the NRe to dCA1 pyramidal cells [36], the key recordings in that study appear to have been in the subiculum and/or prosubiculum areas of the HPC rather than adjacent CA1 (see Fig 1 of this study—the only figure showing recording location). However, other published studies would provide support for our findings. An ultrastructural study reported that synapses from thalamic axons in CA1 make significantly fewer contacts per dendrite than those from entorhinal axons [37], supporting our electrophysiological data and the hypothesis that NRe input to HPC serves to modulate, rather than drive, hippocampal activity, perhaps via inhibitory tone in SL-M. This ultrastructural study found projections from NRe onto spiny dendrites and thus concluded that these contacts were onto CA1 pyramidal cells since GABAergic interneurons are typically considered aspiny. However, recent work has demonstrated the existence of a population of spiny parvalbumin-expressing interneurons in CA1 that have dendrites in SL-M [38], which may have been the target of these NRe neurons. Another recent study reported close proximity of NRe axons onto PSD95-expressing dendritic spines in CA1, again concluding that these projections were onto pyramidal cells [39]. Again, while PSD95 is typically considered a marker of glutamatergic neurons, PSD95 has long been known to also exist on GABAergic neurons, particularly Erb4-expressing parvalbumin cells, as found by ourselves and others [e.g., 40–42].

A recent article from Leprince and colleagues [43] confirmed our main finding using a combination of *in vivo* and *ex vivo* approaches: they found that EC inputs strongly drive CA1 network activity in early postnatal circuitry and that, while NRe stimulation could drive network activity in CA1, this is almost entirely through inhibitory mechanisms. Indeed, this study reported that only 2 out of 26 CA1 pyramidal cells received input from the NRe and it is unclear whether this connection persists into adulthood, with the authors speculating that it may be a transitory circuit in development that helps 'wire' the developing brain. This phenomenon has been widely observed in cortico-thalamic circuits, so this hypothesis is not without precedent (reviewed by [44]). Another recent *in vivo* study found that optogenetic inhibition of NRe projections to CA1 *increases* pyramidal cell activity [45], providing strong evidence to support our view of NRe's principal overall effect on hippocampal activity is one of inhibition and not excitation, even if some sparse monosynaptic projections to pyramidal cells persist into adulthood.

While our electrophysiology data in both mouse and rat would suggest that there are no direct projections from the NRe to CA1, it is possible that sparse, weak synapses could exist but, due to dendritic filtering, produce no detectable response from a somatic whole-cell patch-clamp recording. We thus carried out rabies tracing using pseudotyped SAD B19 viruses with injections made into both dorsal and vCA1 and also the dorsal subiculum to trace inputs specifically from CA1 pyramidal cells. We failed to find evidence of a direct monosynaptic projection from the NRe to CA1 pyramidal cells in either dorsal or vCA1. The possibility remains that the rabies virus will not retrogradely label all axon types with the same efficiency, producing a false negative. However, our control data revealed retrograde labeling of NRe neurons when the injection site included the ventral subiculum, and we also found retrogradely-labeled glutamatergic neurons in regions such as the EC. While our data cannot entirely exclude the existence of a sparse projection from the NRe to CA1 pyramidal cells, if these projections do exist then they are likely to be of little functional significance.

## Circuit implications of the reuniens to HPC projection

*In vivo* electrophysiological studies looking at hippocampal LFPs found that only direct EC stimulation could drive a population spike in the HPC, while NRe activation primarily evoked an LFP in SL-M [19,23], with the latter paper from this group arguing that low frequency stimulation of the NRe likely promotes EC—HPC communication. These papers found that stimulation of both EC and NRe leads to a supralinear summation of LFP amplitude in CA1, leading the authors to reasonably conclude that both regions drive excitation within CA1.

This work and other studies have now shown that the influence of the NRe over CA1 is predominantly to inhibit hippocampal activity by targeting inhibitory interneurons. Neurogliaform cells, with a predominantly NMDA receptor-driven NRe-EPSC ([22] and the present study), will likely have a much stronger influence when HPC is already active. Given that the EC projects to both DG and CA1, and that NRe evokes strong post-synaptic responses in EC neurons, we hypothesize that much of the effect of NRe activation on increasing the magnitude of LFP in CA1 is in fact an effect of feedforward excitation from the EC. There is one report in the literature of midline thalamic stimulation inducing population spikes in CA1 [46], but again, this could be due to feed-forward excitation from the EC.

The NRe projection to CA1, targeting neurogliaform cells, could serve to inhibit distal dendrites of selected CA1 pyramidal cell ensembles and increase the fidelity of EC—HPC communication. Another recent study, using *in vivo* juxtacellular recording in mice, found that neurogliaform cells in CA1 can decouple pyramidal cells from entorhinal-driven gamma oscillations without altering the firing rates of pyramidal cells [47]. Given that the NRe projections to CA1 are highly selective for neurogliaform cells ([22,43] and the present study), we have previously suggested that the NRe recruitment of neurogliaform cells allows NRe to re-route information flow through the HPC, switching between memory encoding and memory retrieval [48].

Neurogliaform cells have a wide axonal arbor and broadly inhibit pyramidal cell dendrites though a combination of GABA$_A$ and GABA$_B$ receptor-mediated mechanisms acting via volume transmission [49], making them well placed to disengage CA1 pyramidal cells from ongoing entorhinal inputs arriving in SL-M alongside those from NRe. We proposed previously that the reuniens to CA1 neurogliaform projection may allow specific ensembles of pyramidal cells to be 'disconnected' from EC, creating a more permissive environment for them to be driven by CA3 inputs via Schaffer collaterals in stratum radiatum [48]. Lesions to the NRe have been shown to impair consolidation, but not acquisition, of spatial memory in the Morris Water Maze in rats [50], so neurogliaform cells could play a key role in memory consolidation by supressing nonrelevant activity during systems consolidation. Others have shown that the NRe is essential for reconsolidation of an extinct fear memory [51–54]. Again, the NRe projection to neurogliaform cells in CA1 could play a key role in this by increasing the signal-to-noise ratio of the reactivated memory by suppressing ensembles active during similar but different episodic memories. This would be consistent with other reports of the NRe being essential for the formation of long-term (but not short-term) recognition memories [55]. Neurogliaform cells have been shown to evoke a longer-lasting inhibitory current compared with other cell types such as fast-spiking basket cells [56] and acting through volume transmission [49] could make their actions too broad

or slow for selecting specific ensembles, although more recent work using juxtacellular recordings in CA1 suggest that neurogliaform cell can selectively modulate network dynamics without broadly suppressing activity [47].

The PFC projections to the HPC are important for impulse control [57], and lesions of the NRe inhibit this function [58]. A recent suggestion for the function of the prefrontal—reuniens—hippocampal circuit is that it actively suppress ongoing memory retrieval [59], and our study provides a mechanism through which this could occur. Indeed, the large NMDA:AMPA ratio of NRe-EPSCs in neurogliaform cells that we reported here and previously [22] suggest that the inhibitory influence evoked via NRe activation is much more likely to be effective in the context of ongoing network activity. Recent behavioral data suggest the requirement for the NRe in spatial memory retrieval or "online" spatial processing, but not for off-line consolidation or long-term storage [60]. Given that the largest NRe-mediated EPSCs were present in entorhinal and prefrontal cortices, it could be that the NRe suppresses hippocampal activation to facilitate prefrontal—entorhinal communication during associative memory consolidation [6]. Alternatively, an inhibitory influence of the NRe over the HPC has been proposed as the mechanism through which the PFC can prevent retrieval, e.g., during extinction of fear memory [61].

Given the large NMDA receptor-mediated component of NRe-EPSCs onto neurogliaform cells, one is tempted to suggest that the role of NRe could be to increase $Ca^{2+}$ in NGF dendrites to activate a NO-dependent suppression of inhibition [62] that could serve to enhance entorhinal-to-CA1 communication. Future behavioral experiments are required to test the function of NRe to CA1 projections, perhaps by using retrograde viruses to allow specific targeting only of those NRe neurons that project directly to CA1.

## Concluding remarks

We have found that, unique to all hippocampal or cortical regions, pyramidal cells in hippocampal region CA1 do not receive monosynaptic input from the thalamus. Even in parts of the hippocampal formation where principal cells do receive direct input from the NRe, the amplitude of the evoked currents is small and likely to be modulatory in nature. These surprise findings suggest that feedforward inhibition driven by neurogliaform cells may be main mechanism through which the NRe influences hippocampal function. The apparently stronger innervation of both PFC and EC pyramidal cells by the NRe raises interesting questions about the flow of information through NRe (hitherto presumed to be the relaying prefrontal control over HPC). The existence of distinct, parallel streams of information through reuniens remains a possibility, and future research at the circuit and synaptic level should focus on NRe's innervation of prefrontal and entorhinal regions.

## Materials and methods

### Ethics statement

Initial mouse pilot experiments (data included in Figs 1 and 3) were conducted in accordance with the Guide for the Care and Use of Laboratory Animals of the US National Institutes of Health (Eighth Edition) and animal protocols (ASP# 14-045) approved by the animal care and use committee (ACUC) at the National Institute of Child Health and Human Development. All remaining mouse work was conducted in accordance with the UK Animals (Scientific Procedures) Act 1986 after local ethical review by the Animal Welfare and Ethical Review Board at the University of Exeter under project licence PAC082CD3. All rat procedures were performed in accordance with United Kingdom Animals Scientific Procedures Act (1986) and associated guidelines under project licence number PP7058522 and were approved by the University of Bristol Ethical Review Committee. All efforts were made to minimize any suffering and the number of animals used.

### Animals

We used *Nkx2-1-cre* [63]:RCE or *Nkx2-1-cre:*Ai9 and *Htr3a*-GFP [64] mice to target interneurons of MGE or CGE origin, respectively, in electrophysiological experiments. *Nkx2-1-cre* mice were obtained from Jackson laboratories (C57BL/6J-Tg(Nkx2-1-cre)2Sand/J, stock number 008661) and Htr3a-GFP mice (Tg(Htr3a-EGFP)DH30Gsat) were cryo-recovered from MMRRC (NC, USA) and back-crossed onto C57BL/6J mice (Charles River, UK). We used *Emx1-cre* mice [65]

crossed with floxed TVA mice [66] to allow specific targeting of pyramidal cells for monosynaptic rabies tracing. *Emx1-cre mice were obtained from Jackson laboratories (B6.129S2-Emx1tm1(cre)Krj, stock number 005628) and floxed TVA mice (LSL-R26Tva-lacZ) were kindly provided by Prof Dieter Saur (Technical University of Munich, Germany). All animals were maintained on a 12 h constant light/dark cycle and had access to food and water ad libitum and were group-housed wherever possible. We used standard enrichment that included cardboard tubes, wooden chew blocks, and nesting material.*

All rat experiments were carried out in naïve male Lister Hooded rats (Envigo, UK) weighing 300–450 g at the start of the experiments. Animals were housed in groups of 2–4, under a 12 h light/dark cycle (light phase, 8.00 PM to 8.00 AM) with *ad libitum* access to food and water. Sacrifice for *ex vivo* slices occurred 2–3 h into the dark cycle.

## Drugs and chemicals

CGP55845 (Catalogue # 1248), DNQX (Catalogue # 2312/10), D-AP5 (Catalogue # 0106/1), and picrotoxin (Catalogue # 1128) were purchased from Tocris Bioscience, and all other chemicals were purchased from Sigma-Aldrich unless otherwise stated.

## Stereotaxic injections for electrophysiology experiments: mice

For optogenetic experiments, we used *Nkx2-1-cre*:Ai9, *Nkx2-1-cre*:RCE, or *Htr3a*-GFP mice of both sexes, totaling at 65 mice, with the age at the time of stereotaxic injection ranging from 2 to 7 months. Mice of both sexes were used for stereotaxic surgery. Mean weight of the mouse prior to stereotaxic surgery was 26 g (ranging from 17.5 to 49.5 g). Two different viruses were used for stereotaxic surgeries: AAV8-hSyn-Chrimson-TdTom (UNC Viral Vector Core, USA, contributed by Ed Boyden; titer $6.3 \times 10^{12}$ viral particles/ml). AAV8-hSyn-Chronos-GFP (UNC Viral Vector Core, USA. contributed by Ed Boyden; titer $3.1 \times 10^{13}$ viral particles/ml).

For the surgery, the mice were anesthetized with 5% isoflurane and anesthesia was maintained with use of 1.5 to 2.5% isoflurane (flow rate of ~2 L min$^{-1}$ O$_2$). The mice were placed on a heated pad (37 °C) for the duration of the surgery and given 0.1 mg/kg of buprenorphine (buprenorphine hydrochloride, Henry Schein) subcutaneously at the start of surgery as an adjunct analgesic, plus carprofen 1 mg/kg (Rimadyl, Henry Schein) was given at a dose of 5 mg/kg subcutaneously post-surgery and on subsequent days, as required. To target NRe, we used the following coordinates: A/P −0.8 mm, M/L 0.0 mm, D/V 3.8 mm from pia, and with 300 nl of virus (infused at 100 nl min$^{-1}$). After the surgery, the mice were allowed at least a 3-week recovery period to allow sufficient time for the expression of the viral construct. For whole-cell patch clamping experiments, AAV8-hSyn-Chronos-GFP or AAV8-hSyn-Chrimson-TdTom were used for *Nkx2.1*-cre*:Ai9 or *Htr3a*-GFP mice, respectively, although a small number of *Htr3a*-GFP mice received AAV8-hSyn-Chronos-GFP to allow direct comparison of EPSC amplitude in the same population of neurons.

## Stereotaxic injections for electrophysiology experiments: rats

Each rat was anesthetized with isoflurane (4% induction, 2.5%–3.5% maintenance) and secured in a stereotaxic frame with the incisor bar set 3.3 mm below the interaural line. Eye drops (0.1% sodium hyaluronate; Hycosan, UK) were applied and body temperature was maintained at 37 °C using a homeothermic heat blanket (Harvard Apparatus, USA). The scalp was further anesthetized using topical lidocaine (5% m/m; TEVA; UK) and disinfected with chlorhexidine, cut, and retracted. Bilateral craniotomies were made using a burr at the following coordinates with respect to Bregma: anterior-posterior (AP)—2.0 mm, mediolateral (ML) ± 1.4 mm. Virus was injected bilaterally into NRe via a 33-gauge 12° beveled needle (Esslab) attached to a 5 μl Hamilton syringe which was mounted at a 10° angle in the ML plane to avoid the sinus, with the eyelet of the needle facing medially. The needle was lowered 7.5 mm below the surface of the skull measured from the burr hole and 100 nl of virus was delivered via each burr hole at a rate of 200 nl min$^{-1}$, with the needle left in situ for 10 min after each injection. AAV9-CaMKii-hChR2(E123T/T159C)-mCherry (Addgene 35512; $3.3 \times 10^{13}$ genome copies/ml) obtained from University of Pennsylvania Vector Core.

## Monosynaptic retrograde tracing

For anatomical experiments we used the monosynaptic rabies tracing method that has been previously reported by others [31]. Two mouse lines (*Emx1-cre* and floxed TVA) were crossed together in order to ensure that the modified rabies virus only targets the pyramidal cells, with *Emx1-cre* mice used as controls to ensure the rabies virus did not transduce neurons in the absence of TVA. A total of 22 mice of both sexes were used, with 2 mice being excluded from the analysis due to failed injections. The age of the mice used ranged from 2 to 6 months, with pre-surgical weights from 18.9 to 40.6 g (mean age 3.5 months, mean weight 24.6 g). To highlight the efferent projections from NRe to HPC, we elected to inject into dorsal and vCA1 using the following coordinates: dCA1 was targeted at A/P −2 mm (relative to Bregma), M/L −1.5 mm and D/V 1.35 mm (from pia) and vCA1 at A/P −2.8 mm (relative to Bregma), M/L −2.4 mm and D/V 4.2 mm (from pia). The stereotaxic injections of AAV8-FLEX-H2B-GFP-2A-oG (Provided by John Naughton at Salk Institute Viral Vector Core, USA, titer $3.93 \times 10^{12}$ viral particles/ml), followed by injection EnvA G-deleted Rabies-mCherry (Provided by John Naughton at Salk Institute Viral Vector Core, titer $6.13 \times 10^{8}$ viral particles/ml; or from Viral Vector Core facility of the Kavli Institute for Systems Neuroscience, NTNU, Norway, titer $2.6 \times 10^{10}$ viral particles/ml) 2 weeks after the initial viral injection were performed in the right hemisphere only. The mice were maintained for 2 weeks to provide optimal time for expression, and were killed by transcardial perfusion/fixation with 4% paraformaldehyde (Cat number P6148 Sigma-Aldrich, UK) in 0.1 M phosphate buffer (PB).

Following the transcardial perfusion, the brains were dissected out and post-fixed for 24 h in 4% PFA solution, after which they were cryoprotected using the 30% sucrose in PBS solution (Catalogue # P4417). Once cryoprotected, the brains were sliced at 50 µm using the freezing microtome (Leica, SM2010 R). Selected slices (1 in 5 serially, increasing to 1 in 3 between −0.5 and −1.8 Bregma to ensure thorough representation of NRe of the thalamus across the AP axes) were mounted using the Hard Set mounting medium with DAPI (Vectashield, Vector Lab, Catalogue # H-1500-10) and the fluorescent fibers were visualized with CoolLED on Nikon Eclipse E800, using 4× objective. Representative photos of projection patterns can be found in Figs 1 and S1.

## Slice preparation and electrophysiology

Mice of age 2 months and above were used for stereotaxic surgeries. A minimum of 3 weeks recovery period following the stereotaxic surgery was allowed. Mice were anesthetised with isoflurane and the brain was rapidly dissected out in room temperature NMDG cutting solution, containing (in mM): 135 NMDG (Catalogue # M2004), 20 Choline bicarbonate (Catalogue # C7519), 10 glucose (Catalogue # G7021), 1.5 $MgCl_2$ (Catalogue # M9297), 1.2 $KH_2PO_4$ (Catalogue # P0662), 1 KCl (Catalogue # P5405), 0.5 $CaCl_2$ (Catalogue # C5670), and saturated with 95% $O_2$ and 5% $CO_2$ (pH 7.3–7.4). Coronal or horizontal slices (400 µm) were cut to target PFC and dCA1 versus vCA1, subiculum, and EC, respectively, using a VT-1200S vibratome (Leica Microsystems, Germany). Afterwards, the slices were transferred into a chamber containing recording aCSF, composed of (in mM): 130 NaCl (Catalogue # S5886), 24 $NaHCO_3$ (Catalogue # S6014), 3.5 KCl, 1.25 $NaH_2PO_4$, 2.5 $CaCl_2$, 1.5 $MgCl_2$, and 10 glucose, saturated with 95% $O_2$ and 5% $CO_2$ (pH 7.4) and placed in a water bath at 37 °C for 30 min, following which they were kept at room temperature until recording.

For rat experiments, after a minimum of 10 days following viral injection, animals were anesthetized with 4% isoflurane and decapitated. Brains were rapidly removed and placed into ice-cold sucrose solution (in mM: 189 sucrose, 26 $NaHCO_3$, 10 D-glucose, 5 $MgSO_4$, 3 KCl, 1.25 $NaH_2PO_4$, and 0.2 $CaCl_2$) bubbled with 95% $O_2$/5% $CO_2$. Dorsal hippocampal recordings were made from parasagittal or coronal slices and ventral HPC from horizontal slices, cut at 350 µm thickness using a vibratome (7000smz-2, Camden Instruments), before incubation at 34 °C for 1-h after dissection in a slice holding chamber filled with artificial cerebrospinal fluid (aCSF, in mM: 124 NaCl, 26 $NaHCO_3$, 10 D-glucose, 3 KCl, 2 $CaCl_2$, 1.25 $NaH2PO_4$, and 1 $MgSO_4$). Slices were subsequently stored at room temperature until use.

For recordings, individual mouse brain slices were attached onto 0.1% poly-L-lysine (Sigma-Aldrich, Catalogue # P8920) coated glass-slides and placed into the upright microscope and visualized using infrared differential interference contrast microscopy (Olympus BX51 or Scientifica SliceScope). CoolLED pE-4000 system was used to visualize the fibers

as well as interneurons, and to provide optogenetic stimulation. The slices were submerged in recording aCSF, warmed to 32–34 °C, and the rate of perfusion was kept at 5 ml/min. The recording electrodes were typically 3–5 MΩ size and were pulled from borosilicate glass (World Precision Instruments). The intracellular solution used had the following composition (in mM): 135 Cs-methanesulfonate (Catalogue # C1426), 8 NaCl, 10 HEPES (Catalogue # H3375), 0.5 EGTA (Catalogue # E3889), 4 MgATP (Catalogue # A9187), 0.3 Na-GTP Catalogue # (G8877), 5 QX314 (Catalogue # 552233), plus 2 mg/ml biocytin (VWR International, UK), and at pH 7.25 adjusted with CsOH and 285 mOsm.

For rat electrophysiological recordings, slices at were placed in a submerged recording chamber and perfused with 34 °C aCSF at ~2 ml min$^{-1}$. A stimulating electrode (FH-Co, CBABAP50, USA) was placed in SL-M. Recordings were made from neurons in stratum pyramidale or lacunosum-moleculare, targetted under infra-red illumination with an upright microscope and whole-cell patch clamped using 2–6 MΩ boroscillicate glass electrodes (GC150-10F, Harvard Apparatus) filled with potassium gluconate internal (in mM: 120 k-gluconate, 40 HEPES, 10 KCl, 2 NaCl, 2 MgATP, 1 MgCl, 0.3 NaGTP, 0.2 EGTA and either 0.1 Alexa-594 hydrazide or 2.7 biocytin, pH 7.25, and 285 mOsm). Recordings were obtained using a Molecular Devices Multiclamp 700B, filtered at 4 kHz and digitized at a sample frequency ≥20 kHz with WinLTP2.30 [67] or pClamp10 software. Resting membrane potential (RMP) was recorded immediately after entering the whole-cell configuration, thereafter neurons were kept at −70 mV by injection of constant current. Intrinsic membrane properties were recorded by injection of square-wave hyperpolarising and depolarizing currents. Optogenetic stimulation was applied with a 470 nm LED (M470L3, Thorlabs) triggered by 2 ms TTL pulses sent to an LEDD1B driver (Thorlabs) via a 40× immersion objective (Olympus or Nikon). In all experiments transduction of NRe was confirmed by recording of photosensitive cells in hippocampal and/or medial PFC slices [30]. Cell morphology was confirmed using either fluorescent visualization of Alexa594 signal, or post-hoc biocytin staining.

For mouse optogenetic experiments, a train stimulation with 5 pulses of 470 nm or 660 nm was used to excite the Chronos or Chrimson opsins, respectively. The presence or absence of responses was recorded in voltage clamp mode. Cells that were found to have a response to a train stimulation were then switched onto repeated single pulse protocol (ISI of 10 s), and the AMPA response was recorded at a holding potential of −70 mV. GABA-R antagonists were bath applied from the start in HPC and subiculum, but not in EC or PFC due to epileptiform activity being observed upon NRe stimulation with GABA-R antagonists present. The extracellular GABA$_A$ and GABA$_B$ receptor antagonists used were picrotoxin (100 μM) and CGP55845 (1 μM). 10 μm of DNQX was added to abolish the AMPA current at −70 mV, after which the cell was switched to +40 mV to record the NMDA current. To confirm the identity of NMDA current, D-AP 5 (100 μM) was added at the end of the recording. Whole-cell patch-clamp recordings were made using a Multiclamp 700A or 700B amplifier (Molecular Devices, Sunnyvale, CA, USA). Signals were filtered at 3 kHz and digitized at 10 kHz using a Digidata 1322A or 1440A and pClamp 9.2 or 10.2 (Molecular Devices, USA). Recordings were not corrected for a liquid junction potential. The recordings were then imported into IgorPro (Wavemetrics, OR) using Neuromatic (Thinkrandom, UK) for further analysis.

Total number and age range of mice (on experimental day) used for whole-cell patch clamping data was: 28 mice of both sexes aged 4–7 months old for hippocampal formation (Figs 1 and 2), 15 mice of both sexes aged 4–7 months old for EC (Fig 5) and 21 mice of both sexes aged 4–8 months old for PFC (Fig 4).

### Post hoc morphological recovery

Mouse tissue slices were post-fixed in 4% PFA solution for an hour after patch clamping recordings and then transferred to PBS (Melford, UK). Slices were washed thrice in 0.1 M PBS solution, followed by three washes in PBS with 0.5% Triton X (Sigma, UK). Two blocking steps were used: 100 mM of Glycine (Sigma, UK) incubation for 20 min, and then 1-h long incubation with blocking buffer with 5% goat serum (VectorLabs, UK) at room temperature. Incubation with streptavidin (1:500 dilution, VectorLabs, UK) was done for 2 h at RT in a carrier solution consisting of 5% goat serum and PBS. Finally, the slices were washed in PBS and transferred to 30% sucrose (Thermofisher, UK) until cryoprotected. Then slices were re-sectioned at 100-μm thickness using SM2010R freezing microtome (Leica, UK) at −20 °C and mounted onto glass-slides and HardSet mounting medium with DAPI was used (VectorLabs, UK).

For rat experiments, following recording, slices were fixed in 4% paraformaldehyde overnight and subsequently stored in 0.1 M PB until staining. Slices were washed 6 × 10 min in PB, then incubated in 3% $H_2O_2$ in PB for 30 min to block endogenous peroxidase activity, then washed again as above before incubation in 1% (vol/vol) avidin–biotinylated HRP complex (ABC) in PB containing 0.1% (vol/vol) Triton X-100 at RT for 3 h. Slices were then washed as above before incubation in DAB solution (1× gold + 1× silver tablet in 15 mL distilled water) for 5–10 min until staining of neuronal structures was visible. The reaction was stopped by transferring slices to cold PB, followed by further washing as above. Slices were mounted, cover slipped using Mowiol mounting media, and allowed to dry overnight before imaging.

## Data analysis

For quality control, cells with changes in input resistance of over 20% throughout the course of the experiment were excluded from the data analysis. The AMPA receptor-mediated EPSC was determined as the maximal EPSC peak at −70 mV and NMDA receptor-mediated EPSC as the highest peak at +40 mV. GraphPad Prism (GraphPad, CA) was used for statistical analysis. Data were tested for normality using the D'Agostino and Pearson test and subsequently analyzed by parametric or nonparametric tests as appropriate. Unless otherwise stated, all values are mean ± SEM. RNAseq data from [32] were downloaded from NCBI Gene Expression Omnibus [68], accession GSE67403) and analyzed in R Statistical Software v4.3.1 [69] and RStudio v 2023.6.1.524 [70], using the tidyverse [71] and ggplot2 packages [72].

For rat experiments, traces from intrinsic electrophysiological experiments were imported into MATLAB and analyzed using custom written code. Synaptic responses were analyzed using MATLAB or WinLTP software [67].

## Supporting information

**S1 Fig. NRe axon distribution in coronal and horizontal sections. A**, coronal section, showing injection site, **B**, more caudal section, showing the fibers in dorsal hippocampus, localized to S-LM. **C** and **D**, show hatched areas from B, on a higher magnification. **E**, horizontal slice, showing dense projections from nucleus reuniens in ventral prefrontal cortex, ventral CA1 and subiculum (**F**) and entorhinal cortex (**G**).
(TIFF)

**S2 Fig. No difference in NRe-EPSC between Chronos and Chrimson, in CGE-derived neurogliaform cells in CA1. A**, representative image of Chronos-GFP NRe fibers (green) in a *Htr3a*-GFP mouse. **B**, representative image of Chrimson-TdTom NRe fibers (red) in a *Htr3a*-GFP mouse. **C**, representative traces for NRe-EPSCs (AMPA-mediated) in CA1 CGE-derived neurogliaform cells from NRe axons transduced with either AAV8-hSyn-Chronos-GFP (upper trace) or AAV8-hSyn-Chrimson-TdTom (lower trace). **D**, no significant difference in median peak AMPA-mediated NRe-EPSC current between fibers expressing Chronos or Chrimson (Chronos versus Chrimson: 10.8 (IQR: 4.0–13.7, *n* = 8) versus 11.2 (IQR: 7.3–21.2, *n* = 17); *p* = 0.4143, Mann–Whitney test). These data can be found in the S1 Data.
(TIFF)

**S3 Fig. Comparison of NRe input to MGE vs. CGE-born neurogliaform cells, and between dCA1 and vCA1.** Representative images of **A**, CGE-derived NGF in vCA1 (arrow); **B**, MGE-derived NGF in vCA1; **C**, putative CGE-derived NGF in dCA1 (TdTom −ve in *Nkx2.1*-cre:Ai9 mouse); **D**, MGE-derived in dCA1 (arrow). **E**, AMPA-R mediated NRe did not vary significantly between NGFs of different embryonic origin (CGE *vs.* MGE *vs.* indeterminate origin: 9.12 ± 1.25 pA *vs.* 13.0 ± 3.2pA *vs.* 8.94 ± 2.23pA; *p* = 0.34, Kruskal–Wallis test). Similarly, **F**, NRe-EPSC paired pulse ratio did not vary by embryonic origin (CGE *vs.* mGE *vs.* indeterminate origin: 1.27 ± 0.11 *vs.* 1.03 ± 0.14 *vs.* 0.88 ± 0.09; *p* = 0.39, Kruskal–Wallis test). **G**, mean AMPA NRe-EPSC did not significantly differ between dCA1 and vCA1 NGFs, pooled across embryonic origin (dCA1 *vs.* vCA1, 9.88 ± 1.3 pA *vs.* 14.3 ± 3.7pA; *p* = 0.6504, Mann–Whitney test). **H**, probability of NRe input in neurogliaform cells in dorsal and ventral CA1 was 76.5% and 81.3%, respectively. These data can be found in the S1 Data.
(TIFF)

**S4 Fig.  Optogenetic activation of NRe axons in rat hippocampus (HPC) evokes a post-synaptic response in GAB-Aergic interneurons but not pyramidal cells in either dorsal or ventral CA1 (vCA1). A**, post hoc recovery of CA1 neurons in dorsal HPC of rat. **B,** lack of EPSP for pyramidal cell shown in A in response to optogenetic stimulation (top), plus response to depolarizing and hyperpolarising current steps (bottom). **C**, total connection probabilities for CA1 pyramidal cells and GABAergic neurons in SL-M, with positive responses in gray and null responses in black. Data from 21 rats. **D,** example EPSP from an SLM-interneuron (purple) and a pyramidal cell (black) recorded from vCA1.
(TIFF)

**S5 Fig.  Control injections for retrograde monosynaptic tracing. A**, representative example of injection site for retrograde labeling viruses in *Emx1-cre:*TVA mouse. Both green (transduced by the helper AAV virus) and red (transduced by the pseudotyped rabies virus) cells can be seen. **B**, in a control mouse (*Emx1-cre*) without TVA, the pseudotyped rabies virus was unable to enter CA1 pyramidal cells ($n = 6$ mice). Scale bar represents 100 μm. **C–E**, schematic images showing spread of viral representative spreads of virus for injections in the dCA1 (C), vCA1 (D), and vSub (E) regions. Each figure is a composite of 4 injections, with the darker color indicating more mice. **F and G,** representative example of injection on low (**F**) and high (**G**) magnification of injections that spread between vCA1, prosubiculum, and vSub. **H and I**, retrogradely-labeled neurons were present in NRe only in mice with starter cells located in prosubiculum and subiculum, confirming our observations from anterograde optogenetics experiments (Fig 1).
(TIFF)

**S6 Fig.  *Emx1* is expressed throughout the hippocampus (HPC).** Secondary analysis of Cembrowksi and colleagues, 2016, https://doi.org/10.1016/j.neuron.2015.12.013) revealed that *Emx1* expression is ubiquitous throughout HPC, albeit at (nonsignificantly) lower levels in ventral CA1 compared with dorsal or intermediate CA1. These data can be found in the S1 Data and the code generating the plots and associated analysis can be found in S1 Code.
(TIFF)

**S1 Data.  Numerical data for figures in the manuscript.**
(XLSX)

**S1 Code.  The code generating the plots and associated analysis.**
(R)

## Acknowledgments

We are grateful to Prof John Aggleton (University of Cardiff) for advice and discussions on anatomy. We gratefully acknowledge the support of the Inger and George Simpson Biological Psychiatry Scholarships.

## Author contributions

**Conceptualization:** Lilya Andrianova, Zafar I. Bashir, Michael T. Craig.

**Data curation:** Lilya Andrianova, Paul J. Banks, Michael T. Craig.

**Formal analysis:** Lilya Andrianova, Paul J. Banks, Clair A. Booth, Michael T. Craig.

**Funding acquisition:** Paul J. Banks, Jonathan Cavanagh, Zafar I. Bashir, Chris J. McBain, Michael T. Craig.

**Investigation:** Lilya Andrianova, Paul J. Banks, Clair A. Booth, Erica S. Brady, Gabriella Margetts-Smith, Shivali Kohli, Michael T. Craig.

**Methodology:** Lilya Andrianova, Clair A. Booth, Michael T. Craig.

**Project administration:** Lilya Andrianova, Zafar I. Bashir, Michael T. Craig.

**Software:** Michael T. Craig.

**Supervision:** Zafar I. Bashir, Chris J. McBain, Michael T. Craig.

**Visualization:** Lilya Andrianova, Paul J. Banks, Clair A. Booth, Michael T. Craig.

**Writing – original draft:** Lilya Andrianova, Paul J. Banks, Michael T. Craig.

**Writing – review & editing:** Lilya Andrianova, Paul J. Banks, Clair A. Booth, Erica S. Brady, Gabriella Margetts-Smith, Shivali Kohli, Jonathan Cavanagh, Zafar I. Bashir, Chris J. McBain, Michael T. Craig.

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
