## [Editor Report · Decision Letter 0]

27 Jan 2025

Dear Dr Craig, 

Thank you for submitting your manuscript entitled "Hippocampus does not appear to be a major target of thalamic nucleus reuniens" for consideration as a Research Article by PLOS Biology.

Your manuscript has now been evaluated by the PLOS Biology editorial staff and I am writing to let you know that we would like to send your submission out for external peer review.

Once your full submission is complete, your paper will undergo a series of checks in preparation for peer review. After your manuscript has passed the checks it will be sent out for review. To provide the metadata for your submission, please Login to Editorial Manager (https://www.editorialmanager.com/pbiology) within two working days, i.e. by Jan 29 2025 11:59PM.

Kind regards,

Taylor

Taylor Hart, PhD, 

Associate Editor

PLOS Biology

thart@plos.org

---

## [Decision Letter · Decision Letter 1]

3 Mar 2025

Dear Dr Craig,

Thank you for your patience while your manuscript "Hippocampus does not appear to be a major target of thalamic nucleus reuniens" was peer-reviewed at PLOS Biology. It has now been evaluated by the PLOS Biology editors, an Academic Editor with relevant expertise, and by several independent reviewers. 

In light of the reviews, which you will find at the end of this email, we would like to invite you to revise the work to thoroughly address the reviewers' reports.

As you will see below, the reviewers described the findings as potentially important and surprising. However, they both raised concerns about missing methodological details, lack of clarity in the text and figures, and some aspects of the interpretations. R1 focused on uncertainties in the viral labeling experiments, while R2 focused on uncertainties in the location of the cells used in physiological recordings. R2 also discusses possible explanations for the discrepancy between this work and prior reports, suggesting that they could arise either from differences in anatomical definitions or because not all relevant areas of CA1 were recorded from. These points will need to be resolved before the paper can be considered for publication.

As such, we would like to invite a Major Revision of this submission. The revised manuscript will need to provide the missing details, expand the discussion of implications for circuit function, and modify the claims in the text to be more precise, in line with the reviewer's comments. We also strongly recommend changing the title to something like "Hippocampal pyramidal cells do not appear to be a major target of thalamic nucleus reuniens". 

Given the extent of revision needed, we cannot make a decision about publication until we have seen the revised manuscript and your response to the reviewers' comments. Your revised manuscript is likely to be sent for further evaluation by all or a subset of the reviewers.

**IMPORTANT - SUBMITTING YOUR REVISION**

*Re-submission Checklist*

*Published Peer Review*

*PLOS Data Policy*

*Blot and Gel Data Policy*

Sincerely,

Taylor

Taylor Hart, PhD, 

Associate Editor

PLOS Biology

thart@plos.org

REVIEWS:

Reviewer #1: Review of PBIOLOGY-D-25-00109R: "Hippocampus does not appear to be a major target of thalamic nucleus reuniens" by Andrianova et al.

In this manuscript, Andrianova and colleagues examine the degree to which axons from the nucleus reuniens form synapses on neurons in the hippocampus, prefrontal cortex, and entorhinal cortex. This question is important because the RE-to-CA1 pathway is presumed to be the major pathway linking prefrontal cortical signals to the principal neurons in the hippocampus. Although some past work has found evidence for these connections, Andrianova and colleagues demonstrate that these specific connections appear absent. The authors perform these experiments using optogenetic strategies paired with slice recordings, and with data from monosynaptic rabies mapping. Collectively, the experiments are sound, the results compelling, and the interpretations important. I assume the field will appreciate the clarification about how the RE circuit nodes are connected in distinct ways to different cell types in CA1, EC, and PFC.

After review of the manuscript, I have the following recommendations/comments/concerns:

- The title uses the terminology 'major target', but this seems hard to establish as hippocampal neuogliaform cells (and potentially other interneuron types localized to SLM) appear to consistently receive RE input. I would favor clarifying it with 'Hippocampal pyramidal neurons do not…' (though the subiculum is arguably part of the hippocampus) or better yet by emphasizing the role of inhibition, such as 'Interneurons in CA1 are the major hippocampal target of thalamic nucleus reuniens'

- The experiments by which the authors confirm the lack of input (with interleaved interneuron recordings from the same slices) provides strong evidence that the inputs are at best very sparse and otherwise absent. They should be commended for this approach. The authors discuss previous papers that suggest the presence of these inputs with proper context and treat them quite fairly. As one of those authors, I appreciate that.

- The high probability of activating neurogliaform cells is an important mechanism by which RE might control CA1. These interneurons drive GABA-B mediated inhibition of the distal dendrites, which presumably exerts control over CA1 neurons switching from linear to non-linear forms of synaptic integration. There is some text devoted to this in the discussion, but a more thorough examination of how RE might contribute to information processing in CA1 could improve the manuscript (see more below).

- The rabies results need more methodological unpacking/description in the text, and perhaps a schematic to demonstrate how the experiments were performed (injection sites, viruses, etc.) and which colors correspond to which cells (containing helper-virus, double labeled starter cells, and connected cells). As it stands now, it takes a lot of work for the reader to understand the experiments without these details (perhaps move the C-E from Supp Fig 4 to main?). The images are very small and hard to parse- part of this is the wish to show the whole field, and part is image quality.

- If the rabies results are already published, can the authors reproduce the data in the figures (with the text clearly stating this) to allow the reader to gauge the efficiency of the rabies labeling? Specifically, it would be nice to know the number of starter cells in CA1 relative to monosynaptic rabies labeled cells in CA3 and EC. If there are 10 starters cells and 10 EC cells, then the reader would know the labeling efficiency is very low and be able to place the lack of RE cells in context. Is this the SAD strain of psuedotyped rabies or the CVS strain? These two are very different in terms of transcellular efficiency. The authors should also discuss the possibility that rabies virus particles may not retrogradely label different axons with the same efficiency- axonal tropism for modified rabies is a relatively unexplored issue.

- In the concluding paragraph, I think describing the RE input patterns (or the RE-CA1 pathway) as a 'red herring' is taking it too far. The authors have not demonstrated that this pathway is unimportant for hippocampal function; in contrast, they have identified the key cellular targets of RE in CA1. Without devaluing the finding that CA1 pyramidal cells are not receiving RE inputs, such cellular targeting to interneurons selectively in CA1 may permit RE to differentially influence processing by reducing CA1 output yet increasing that of subiculum. I think more discussion on how RE-to-CA1 neurogliaform cells impact information processing might come across as a greater advance for the field.

- The presentation of the text and figures could be improved. For example, I believe the first sentence under the heading "Implications for NRe function" should read "hippocampal projections to the prefrontal cortex" rather than "Prefrontal cortical projections to the hippocampus", right? Figure 4 axis legends should read "AMPA" rather than "AMAP". Figure 4 and 5 have the same title, but are focused on two different regions. The pie charts are repeated across figures, and the figure organization could be improved such that Figure 3 is not referred to in the text before Figure 2.

-----

Reviewer #2: In the current manuscript Andrianova and coauthors analyze the projections between the nucleus reuniens (NRe) and hippocampus. To this end they integrated ex vivo optogenetics and electrophysiology, alongside monosynaptic circuit-tracing. Based on the performed experiments they came to very surprising and exciting (if true) conclusion that the prefrontal and entorhinal cortices, subiculum and prosubiculum, but not the CA1 pyramidal neurons, receive the direct input from NRe. Trying to understand the source of discrepancy between this work and former studies which identyfied synapses between NRe and pyramidal CA1 neurons (Wouterlood et al., 1990; Herkenham, 1978; Kajiwara et al., 2008; Vertes et al., 2006) I came to conclusions that the authors should possibly more precisely define what they studied.

1. One of the source of the controversy can be definition of the prosubiculum and subiculum areas. For example, The Mouse Brain in Stereotaxic Coordinates by Paxinos and Franklin does not recognizes prosubiculum. It is just CA1 area. Clear introduction of the prosibiculum area and a precise graphic illustration how this area was defined by the authors (both in the coronal and sagittal planes) in each experiment would be useful. For example, the areas labeled by the authors as vSub and ProSub (supplementary Fig. 4E) are named as CA1 area by Paxinos and Franklin (Figure 60, Bregma -3.52 mm), although Sub (S) is distinguished in this atlas.

2. The authors should precisely describe what parts of dCA1 and vCA1 were probed with patch clamp analysis. Judging from the pictures included in the manuscript, I am not sure if they analyzed more anterior parts of dCA1 (Bregma > -2.3 mm). This is an important issue firstly because dCA1 neurons differ functionally in the anterior-posterior axis and secondly because a recent study (https://doi.org/10.7554/eLife.101736.3 ) showed NRe axons in close proximity to dCA1 excitatory synapses of the pyramidal neurons. However, in this work the authors focused on the anterior dCA1 (AP, Bregma from -1.7 to 2.06). This discrepancy with current data should be discussed.

3. The information on number, age and sex of the animals in different experiments is missing. The authors just indicate that the age of animals was between 2 (when the hippocampus may be not fully developed) and 7 months.

Minor issues:

Title: The authors would like to conclude that Hippocampus does not appear to be a major target of thalamic nucleus reuniens. However, based on theirs current and published data neurogliaform cells in the hippocampus are the major target of NRe.

Fig 2. (A) Please, mark boundaries between prosubiculum, subiculum and Ca1.

(C) and (F) Please, label the neurons on the microphotographs. Especially, on the microphotograph F it is not clear which pyramidal neuron is in CA1 and which in prosubiculum.

Fig. 3. Please, clarify what is n value - cell or animal?

(C) This is unclear why the authors use 2-way ANOVA to analyze data demonstrated on this graph. What are two factors possibly contributing to AMPA currents here?

Fig. 4. Data shown in panel A.iv seem to a replicate of what is shown in Fig. 3A.

The information how many animals were tested, their age and sex is missing.

Fig. 6. The punctuation marks in the legend are confusing. As opposite to earlier legends, here the names of panels (letters) are after the pane description. The text has commas instead of periods.

(J) Please, show microphotographs with higher magnification so the morphology of NRe neurons could be clearly seen.

Materials and methods. 

- Catalog numbers of used chemical are missing. 

- I believe that the authors used Nikkon Eclipse E800, not Nikkon 800 microscope. Please, confirm and correct. Please, indicate also what objectives were used.

The font at the scale bars in supplementary figures 1 is too small to read and the scale bars are too slim.

---

## [Editor Report · Decision Letter 2]

3 Sep 2025

Dear Dr Craig,

Thank you for your patience while we considered your revised manuscript "Thalamic nucleus reuniens preferentially targets inhibitory interneurons over pyramidal cells in hippocampal CA1 region" for publication as a Research Article at PLOS Biology. This revised version of your manuscript has been evaluated by the PLOS Biology editors and the Academic Editor. Your paper encountered a handling delay before the editors were made aware of your revision. We apologize for the additional time that it took to evaluate your revised manuscript.

Based on our Academic Editor's assessment of your revision, we are likely to accept this manuscript for publication. Please also make sure to address the following data and other policy-related requests.

IMPORTANT: Please ensure that your revised manuscript addresses the following bulleted points:

------

**Title

Thank you for revising the title of your paper in response to the reviews. After discussing your paper further within the editorial team, we would recommend some small additional tweaks. Would either of these similar versions work for you?

1. "Hippocampal pyramidal cells of the CA1 region are not a major target of the thalamic nucleus reuniens"

2. "The thalamic nucleus reuniens preferentially targets inhibitory interneurons rather than pyramidal cells in the hippocampal CA1 region"

**Ethics: 

As your ethics statement says that experiments were conducted under the NIH protocols *or* under the UK protocols, please clarify how this was done, and provide additional information on the NIH aspects if applicable (IACUC/ethics committee; protocol; specific guidelines). 

Please make your Ethics statement a separate and independent subheading in the Materials and Methods section.

**Data:

Please supply the numerical values either in the a supplementary excel file or as a permanent DOI’d deposition for the following figures:

3BCDE 

4BCD 

5EFG 

6NOP 

S2D 

S3EFG 

S6

Please cite the location of the data clearly in all relevant main and supplementary Figure legends, e.g. “The data underlying this Figure can be found in S1 Data” or “The data underlying this Figure can be found in https://doi.org/10.5281/zenodo.XXXXX”

Please ensure that you are using best practice for statistical reporting and data presentation. These are our guidelines https://journals.plos.org/plosbiology/s/best-practices-in-research-reporting#loc-statistical-reporting and a useful resource on data presentation https://journals.plos.org/plosbiology/article?id=10.1371/journal.pbio.1002128

If you are reporting experiments where n ≤ 5, please plot each individual data point.

Supplementary files (e.g., excel). Please ensure that all data files are uploaded as 'Supporting Information' and are invariably referred to (in the manuscript, figure legends, and the Description field when uploading your files) using the following format verbatim: S1 Data, S2 Data, etc. Multiple panels of a single or even several figures can be included as multiple sheets in one excel file that is saved using exactly the following convention: S1_Data.xlsx (using an underscore).

**Code availability:

**Abstract

Please note that per journal policy, the model system/species studied should be clearly stated in the abstract of your manuscript. 

Please also add the missing articles (eg '*The* nucleus reuniens (NRe) is a midline thalamic nucleus...).

------------

We expect to receive your revised manuscript within two weeks. 

*Published Peer Review History*

*Press*

Sincerely,

Taylor

Taylor Hart, PhD, 

Associate Editor

thart@plos.org

PLOS Biology

---

## [Editor Report · Decision Letter 3]

16 Sep 2025

Dear Dr Craig,

Thank you for the submission of your revised Research Article "Hippocampal pyramidal cells of the CA1 region are not a major target of the thalamic nucleus reuniens" for publication in PLOS Biology. On behalf of my colleagues and the Academic Editor, Jozsef Csicsvari, I am pleased to say that we can in principle accept your manuscript for publication, provided you address any remaining formatting and reporting issues. These will be detailed in an email you should receive within 2-3 business days from our colleagues in the journal operations team; no action is required from you until then. Please note that we will not be able to formally accept your manuscript and schedule it for publication until you have completed any requested changes.

PRESS

Sincerely, 

Taylor Hart, PhD, 

Associate Editor

PLOS Biology

thart@plos.org